# The Role of Preoperative Chronic Statin Therapy in Heart Transplant Receipts—A Retrospective Single-Center Cohort Study

**DOI:** 10.3390/ijerph20043471

**Published:** 2023-02-16

**Authors:** Dragos-Florin Baba, Horatiu Suciu, Calin Avram, Alina Danilesco, Diana Andreea Moldovan, Radu Catalin Rauta, Laurentiu Huma, Ileana Anca Sin

**Affiliations:** 1Emergency Institute for Cardiovascular Diseases and Transplantation, 540142 Targu Mures, Romania; 2Department of Cell and Molecular Biology, George Emil Palade University of Medicine, Pharmacy, Science and Technology of Târgu Mureș, 540142 Targu Mures, Romania; 3Department of Surgery, George Emil Palade University of Medicine, Pharmacy, Science and Technology of Târgu Mureș, 540142 Targu Mures, Romania; 4Department of Medical Informatics and Biostatistics, George Emil Palade University of Medicine, Pharmacy, Science and Technology of Târgu Mureș, 540142 Targu Mures, Romania; 5Faculty of Medicine, George Emil Palade University of Medicine, Pharmacy, Science and Technology of Târgu Mureș, 540142 Targu Mures, Romania

**Keywords:** statin, atorvastatin, C-reactive protein, heart transplant, complications

## Abstract

Background: Statin therapy has been proven to reduce the risk of cardiovascular events. The objective of our retrospective study was to investigate the relationship between preoperative chronic administration of statins to postoperative 2-month heart transplantation complications. Methods: A total number of 38 heart transplantation recipients from the Cardiovascular and Transplant Emergency Institute of Târgu Mureș between May 2014 and January 2021 were included in our study. Results: In logistic regression, we found a statistical significance between statin treatment and the presence of postoperative complications of any cause (OR: 0.06, 95% CI: 0.008–0.56; *p* = 0.0128), simultaneously presenting an elevated risk for early-postoperative acute kidney injury (AKI). From the statin group, atorvastatin therapy had a higher risk of type 2 diabetes mellitus (T2DM) development (OR: 29.73, 95% CI: 1.19–741.76; *p* = 0.0387) and AKI (OR: 29.73, 95% CI: 1.19–741.76; *p* = 0.0387). C-reactive protein (CRP), total cholesterol (TC), and low-density lipoprotein cholesterol (LDL-c) represented risk factors, atorvastatin administration being independently associated with lower CRP values. Conclusions: Chronic previous administration of statins represented a protective factor to the development of 2-month postoperative complications of any cause in heart transplant receipts.

## 1. Introduction

Heart transplant, after more than 50 years since first human-to-human procedure, remains the gold-standard treatment for patients with end-stage heart failure (HF) [1,2]. The transition from an experimental surgery to a lifesaving procedure has increased the number of patients waiting for heart transplant. In addition, the number of donors does not reach the number of heart transplant candidates. Recently, the use of donor hearts diagnosed with hepatitis C virus infection, left ventricular dysfunction, older age, or history of cocaine use have been included, in order to raise the number of donor hearts [3].

According to the European Society of Cardiology’s 2021 guidelines for heart failure, the first line medical therapy recommended for the treatment of heart failure with reduced ejection fraction (HFrEF) is represented by the angiotensin receptor-neprilysin inhibitor (ARNI) or angiotensin-converting enzyme inhibitor (ACE-I), beta-blocking agent, mineralocorticoid receptor antagonist (MRA), and the new recommendation of sodium-glucose co-transporter 2 (SGLT2) inhibitors such as dapagliflozin and empagliflozin unless the drugs are contraindicated or not tolerated [4,5,6]. Considering the first agents, ARNI is the preferred combination in comparison to ACE-I [4].

Statins, also known as 3-hydroxy-3-methylglutaryl coenzyme A (HMG-CoA) reductase inhibitors, have been demonstrated in decreasing the mortality and morbidity in adults with coronary heart disease [7]. In HF patients, statins did not reduce sudden cardiac death events, but was associated with a decreased rate of hospitalization for worsening HF [8].

However, a meta-analysis of ten randomized controlled trials showed that atorvastatin significantly decreased all-cause mortality, hospitalization for worsening HF, and led to a significant improvement in the left ventricle ejection fraction (LVEF). On the other hand, these benefits could not be established in the same manner in subjects that received rosuvastatin [9]. 

In heart transplant receipts, a meta-analysis including nine studies that investigated the role of statins showed that there was an improved survival in heart transplant recipients receiving this specific drug therapy postoperatively. Statin administration after heart transplantation may also prevent fatal rejection episodes, decrease terminal cancer risk, and reduce the incidence of coronary vasculopathy [10]. The evidence suggests that after heart transplantation, there is an impaired lipid profile, and statin administration should be part of post-transplantation therapy [11].

A retrospective study that included patients undergoing non-cardiac vascular surgery such as carotid endarterectomy, aortic surgery, or peripheral lower extremity revascularization reported that preoperative use of statins decreased cardiovascular postoperative complications [12]. Furthermore, atorvastatin administration before vascular surgery was significantly associated in the first 6 months with the risk reduction of postoperative major adverse cardiovascular events including death from cardiac causes, nonfatal acute myocardial infarction, ischemic stroke, and unstable angina [13]. In contrast, regarding cardiac surgery, a systematic review showed that perioperative statin use was not associated with a decreased incidence in postoperative myocardial infarction, with an elevated risk of acute kidney injury (AKI) [14]. 

Various markers that are routinely determined in the preoperative period have been proven to be prognostic factors in the apparition of postoperative complications such as C-reactive protein (CRP) and low-density lipoprotein cholesterol (LDL-c). In patients undergoing cardiac surgery, high preoperative CRP values showed an increased risk of both early and late mortality [15,16]. In terms of lipid profile, lower circulatory levels of LDL-c represented a protective factor against perioperative myocardial injury in off-pump coronary artery bypass grafting patients [17]. Moreover, elevated LDL-c levels independently predicted long-term cardiac death after coronary artery bypass graft [18]. On the other hand, it has been observed that, during cardiac surgery, the level of plasma lipids falls, having a protective effect on the occurrence of postoperative stroke [19].

Statins represent the main treatment for dyslipidemia, essentially through their acting mechanism on reducing the circulatory levels of LDL-c. Additionally, it has been proven that statin administration decreases the plasma concentrations of CRP in patients with cardiovascular diseases [20]. A previous placebo-controlled trial performed on a large, apparently healthy, cohort showed that rosuvastatin administration in patients with normal baseline LDL-c levels had a 50% reduction rate and lowered the levels of the CRP serum concentrations by 37%, resulting in decreased rates of cardiovascular events [21]. 

The objective of our retrospective study was to investigate the relationship between the preoperative chronic administration of statins, combined with the classic first line HFrEF drug therapy and postoperative 2-month heart transplantation complications. Second, we determined the cut-off of preoperative CRP total cholesterol (TC), respectively, the LDL-c levels, investigating the relationship between these markers and 2-month complications after heart transplant.

## 2. Materials and Methods

### 2.1. Study Design and Patients

This research was performed under the form of a retrospective study, directed from May 2014 to January 2021, at the Emergency Institute for Cardiovascular Diseases and Transplantation, Târgu Mureș, Romania. We examined a total number of 39 patients who underwent heart transplantation at our institution. From the total number of individuals initially enrolled in this study, one was excluded due to insufficient data evidence. All patients completed informed consent forms, the research protocol being validated by the ethics committee of the Cardiovascular and Transplant Emergency Institute of Târgu Mureș, in accordance with the Declaration of Helsinki.

### 2.2. Management and Follow-Up

All of the patients included in this study underwent complete blood analysis, right ventricle biopsies, glucose levels monitoring, periodic 12-lead electrocardiogram, serum creatinine levels, infection screening, and echocardiographic evaluations before and after cardiac transplantation. Therapeutic algorithms were documented and extracted from the previous treatment sheets attached to the patient’s files (Figure 1).

Subjects were divided into two groups based on the presence or absence of chronic statin treatment prior to transplantation. The groups were reported by the means of age, body mass index (BMI), preoperative left ventricle ejection fraction (LVEF), in addition to the lipid profile of the patients, measuring the TC, LDL-c, high-density lipoprotein cholesterol (HDL-c), non-high-density lipoprotein cholesterol (non-HDL-c), very-low-density lipoprotein cholesterol (VLDL-c), and triglyceride (TG) levels. Prognostic scores such as Castelli risk indices I or II, atherogenic index (AI), and atherogenic index of plasma (AIP) were also determined. Castelli risk indices I and II were calculated as LDL-c to HDL-c ratio, respectively, the TC to HDL-c ratio, AI as the non-HDL-c to HDL-c ratio, and AIP as the logarithmic transformation of TG to HDL-c ratio (log_10_[TG/HDL-c]) (Figure 1). 

Statin therapy as well as HFrEF drug combinations were later investigated in relation to the presence of post-procedural complications, taking into consideration the complications of any cause, acute graft rejection, development of newly diagnosed type 2 diabetes mellitus (T2DM), documented paroxysmal atrial fibrillation (AFib), and AKI (Figure 1).

### 2.3. Statistical Analysis

By using MedCalc version 19 for the quantitative data, we determined the mean values, standard deviation (SD), maximum and minimum values. The normality test was evaluated by the Shapiro–Wilk test [22]. We compared the values between the statin group vs. non-statin group using the *t* test for parametric data and the Wilcoxon test for the non-parametric data. We correlated the preoperative elevated marker values with the apparition of complication of any cause, acute graft rejection, postoperative newly diagnosed T2DM, appearance of paroxysmal documented episodes of AFib, and AKI in logistic regression. To evaluate the inflammatory response to different drug associations, CRP levels were evaluated with the presence of complications, determining cut-off values by using receiver operating characteristic (ROC) curves. Similar evaluations were performed with the TC and LDL-c blood levels. The prognostic values of preoperative CRP, TC, and LDL-c were analyzed through logistic regression, taking into consideration the cut-off values determined by the ROC analysis. Correlations were analyzed between the CRP levels and drug therapy by applying the Spearman’s correlation coefficient. In the end, the associations were performed by using the Chi-square (χ^2^) test. The significance threshold was set to 0.05.

## 3. Results

From the 38 patients included in our study, the total group’s mean age was 41.21 years (SD = 13.71). Regarding gender distribution, we counted a total of 34 male receipts (89.5%) and four females (10.5%). A number of 27 subjects did not receive statin therapy (71.1%) compared with 11, who were on chronic statin treatment (28.9%). The mean age of the non-statin group was 38.15 years (SD = 14.74) vs. 48.73 years (SD = 6.60) in the statin group, having lower BMI mean values (22.58; SD = 5.48 vs. 26.84; SD = 2.68) and higher LVEF previous to cardiac transplantation (27.48%; SD = 15.28 vs. 23.64%; SD = 4.03) (Table 1).

There was a statistical difference between the two groups in terms of age (*p* = 0.02) and BMI (*p* = 0.01), both parameters being higher in the statin group. The patient lipid profile revealed mean TC values of 164.37 mg/dL (SD = 46.64) in the non-statin group vs. 169.55 mg/dL (SD = 41.87) in the statin group, mean LDL-c values of 107.22 mg/dL (SD = 34.53) vs. 103.18 mg/dL (SD = 32.31), mean HDL-c values of 34.85 mg/dL (SD = 11.23) vs. 40.45 mg/dL (SD = 7.06), mean non-HDL-c values of 129.52 (SD = 39.87) vs. 129.09 (SD = 41.86), mean VLDL-c values of 20.86 (SD = 10.23) vs. 26.55 (SD = 11.10), and mean TG values of 104.30 mg/dL (SD = 51.16) vs. 132.73 mg/dL (SD = 55.51). Castelli risk index I and II presented mean values of 4.97 (SD = 1.44) in non-Statin group vs. 4.33 (SD = 1.48) in Statin group, respectively, 3.26 (SD = 1.20) vs. 1.65 (SD = 1.10). Regarding AIP, the mean values were 0.46 (SD = 0.21) in the non-statin group compared with 0.49 (SD = 0.20) in the statin group (Table 1). 

From the 11 subjects receiving statin therapy, 72.7% were on atorvastatin (8/11) and 27.3% on rosuvastatin (3/11), with a mean dose of 18.13 mg (SD = 5.30) in the atorvastatin group. Regarding the specific HF chronic therapy, we registered the same number of patients being on spironolactone, having a mean dose of 60.00 mg (SD = 25.93) as well as on beta blockers (78.9%). From the beta blockers, the preferred drug choice was carvedilol (71.1%) with a mean dosage of 19.21 mg (SD = 12.73). A number of 17 patients received ACE-I (44.7%) with a preferred option for ramipril (31.6%) with a mean dosage of 5.42 mg (SD = 2.34) (Table 1).

A proportion of 21.1% from the patients had been previously diagnosed with permanent AFib, 18.5% in the non-statin group, and 27.3% in the statin group. The same numbers and ratios were seen in patients with previous CRT-D implantation (Table 1).

The proportion of complications regardless of cause in our cohort was 78.9%; 88.9% occurring in the non-statin group (24/27) compared with 54.5% in the statin group (6/11). Regarding the specific types of postoperative complications, we counted 18.4% of cases developing acute graft rejection, 21.1% T2DM, 15.8% paroxysmal documented AFib, and 31.6% early-postoperative AKI. There were higher rates of T2DM, and AKI in the statin group compared with the non-statin group (27.3% vs. 18.5%, 54.5% vs. 22.2%, respectively). We observed that all patients that developed T2DM were on atorvastatin from the statin group (Table 2). Even though we found a relatively high rate of early-postoperative AKI, only one patient was discharged with impaired renal function. Other complications that we counted were postoperative infections, need of hemodiafiltration, and moderate to large pericardial effusion (>10 mm). One patient presented cardiogenic shock 2 days after heart transplantation. The cardiogenic shock was also included as one of the complications in our study. All of these events were counted as “complications of any cause”. 

By including the drug therapy in logistic regression, we found that statin treatment was significantly associated with the presence of postoperative complications of any cause (OR: 0.06, 95% CI: 0.008–0.56; *p* = 0.0128) being a protective factor, together with atorvastatin therapy (OR: 0.02, 95% CI: 0.001–0.46; *p* = 0.0132) (Appendix A). Second, the atorvastatin group had a statistically significant risk of T2DM (OR: 29.73, 95% CI: 1.19–741.76; *p* = 0.0387) as well as early-postoperative AKI development (OR: 17.00, 95% CI: 1.26–228.62; *p* = 0.0326). The highest risk of T2DM was seen in patients with atorvastatin and spironolactone combination, with the same odds ratios as statin combined with spironolactone and carvedilol (OR: 29.73, 95% CI: 1.19–741.76; *p* = 0.0387). Ramipril combined with atorvastatin and spironolactone presented the highest risk of AKI (OR: 29.25 95% CI: 1.58–538.60; *p* = 0.0231) (Table 3). 

Another important observation would be the fact that all the patients receiving atorvastatin were on spironolactone as well as statins combined with spironolactone plus carvedilol who were on atorvastatin. Additionally, all the patients with the combination of statins, spironolactone and ramipril were on atorvastatin. 

By using ROC analysis and Youden’s J-point, we determined the optimal cut-off values for preoperative CRP, TC, and LDL-c. The optimal cut-off levels were 2.6 mg/L for preoperative CRP, 153 mg/dL for TC, and 91 mg/dL for LDL-c in the prediction of 2-month apparition of postoperative complications of any cause after heart transplant (Figure 2).

Regarding our determined cut-off levels of CRP, TC, and LDL-c in the appearance of complications regardless of cause, the area under the curve (AUC) showed relatively decreased values for all three markers: 0.633 (95% CI: 0.462–0.783) for CRP, 0.646 (95% CI: 0.474–0.794) for TC, and 0.683 (95% CI: 0.513–0.824) for LDL-c (Table 4).

When looking at the sensitivity and specificity of the markers, CRP > 2.6 mg/L had a sensitivity of 80% with a specificity of 62.5%, TC > 153 mg/dL had a sensitivity of 70% with a specificity of 75%, and LDL-c > 91 mg/dL had a sensitivity of 80% with a specificity of 75% (Table 4).

By applying logistic regression, there was a statistical significant association between CRP (OR: 7.64, 95% CI: 1.24–46.83; *p* = 0.0279), TC (OR: 8.76, 95% CI: 1.24–61.66; *p* = 0.0293), and LDL-c (OR: 11.25, 95% CI: 1.54–82.15; *p* = 0.0169) to the presence of complications, all being prognostic risk factors. We found no statistical association between markers and specific type of complications such as acute graft rejection, new diagnosed T2DM, paroxysmal AFib, and AKI by using these determined cut-off levels (Table 5).

We found a negative correlation between drug therapies and CRP values, but the magnitude of these correlations was small. In particular, atorvastatin administration was inversely correlated with the CRP values (r = −0.30), and using the χ2 test, the statistical association between them was pointed out (*p* = 0.03). Furthermore, we found a statistical association between combinations of statin/atorvastatin with spironolactone and carvedilol with lower blood levels of CRP (Table 6).

## 4. Discussion

In our study, the chronic administration of statins previous to heart transplantation represented a protective factor against 2-month postoperative complications of any cause, while at the same time being a risk factor for the occurrence of early-postoperative AKI. We observed that the addition of spironolactone and ramipril to the statin treatment further increased the risk of AKI. 

From a total number of 11 subjects receiving statin therapy (28.9%), the preferred choice was atorvastatin (21.1%), with a relatively reduced mean dosage (18.13 mg). We noted that the group that had been receiving statin treatment presented statistically significant higher age and BMI. Lower previously determined LVEF, CRP, and LDL-c values were noted in the statin group compared with the non-statin group. 

Previous atorvastatin administration combined with post-transplant immunosuppressive therapy had a significant risk of T2DM development. Various studies have investigated potential diagnostic tools to ensure early detection and prompt treatment of new-onset diabetes after transplantation (NODAT). In renal transplantation, Hecking et al. demonstrated that insulin therapy in the early post-transplant period decreased the chances for NODAT, perhaps because of the insulin-mediated protection of β cells [23]. 

Statin use has been reported to be associated with an increased risk of T2DM, with an elevated relative risk among people without cardiovascular disease [24]. An open-label clinical trial in adults without known atherosclerotic cardiovascular disease or T2DM at the baseline investigated the role of atorvastatin in the physiological mechanism for T2DM risk, proving that high-intensity atorvastatin for 10 weeks increases insulin resistance and insulin secretion [25].

Prior studies have indicated that statin therapy before surgical interventions might be a protective factor against complications, with a favorable postoperative outcome. Perioperative statin exposure in non-cardiac surgery is associated with lower 30-day all-cause mortality, with a reduction in postoperative complications. Additionally, statin use is demonstrated to lower the risk of renal, infectious, and respiratory complications. However, the impact on respiratory and infectious complications remains disputable [26]. 

In patients undergoing cardiac surgery, perioperative statin administration might increase the risk of postoperative AKI [27]. A large trial that included a total number of 1922 patients scheduled for elective cardiac surgery showed no benefit from the perioperative administration of rosuvastatin in the prevention of postoperative atrial fibrillation or perioperative myocardial damage, with patients receiving rosuvastatin being associated with a higher rate of postoperative AKI [28]. In contrast, a meta-analysis including 12 retrospective and three prospective trials reported that preoperative statin therapy had a lower risk of mortality after non-cardiac, cardiac, and vascular surgery [29]. Another meta-analysis noted that atorvastatin, compared with the other statins, was significantly associated with a decreased risk of new-onset AFib in patients undergoing coronary artery bypass grafting [30]. 

Postoperative risk factors have largely been investigated in the medical literature. Elevated markers such as CRP, TC, and LDL-c levels have been reported as being associated with worse prognosis. By using ROC analysis and calculating Youden’s index, cut-off values of 2.6 mg/L for CRP, 153 mg/dL for TC, and 91 mg/dL for LDL-c were determined. In logistic regression, all three markers were significantly associated with the development of postoperative complications regardless of cause, the highest risk being registered for LDL-c levels, presenting a sensitivity of 80% and a specificity of 75%. 

In addition, atorvastatin therapy was inversely correlated with CRP values, underscoring the anti-inflammatory effect of this drug, and potentially through this effect, reducing the subsequent risk of early-postoperative complications. 

The involvement of inflammatory processes in graft transplantation, particularly CRP blood levels, has previously been studied. In heart or renal transplant recipients, high CRP plasma concentrations are associated with allograft failure, independently predicting allograft survival [31,32,33]. CRP has been also associated in cardiovascular diseases such as acute heart failure, where the marked elevation of this marker at patient admission is related to higher all-cause, cardiac, and non-cardiac death rates [34]. 

Regarding the lipid profile, post-transplant dyslipidemia represents a high prevalence complication, secondary to multiple mechanisms including those of immunosuppressive drug-mediated processes [35]. Immunosuppression-mediated hyperlipidemia is characterized by high levels of LDL-c, VLDL-c, and/or total plasma TG. Taking this factor into consideration, it is justifiable that lipid-lowering therapy is routinely used in treatment algorithms of transplant recipients [36]. Wenke et al. reported in a prospective randomized trial that early initiation of simvastatin therapy in heart transplant recipients was associated with significantly better 8-year survival, with a lower incidence of transplant vasculopathy [37]. 

Graft function depends on lifelong immunosuppression therapy. Long-time immunosuppression administration provides protection against organ rejection, but at the same time, it raises the risk of developing opportunistic infections and cancers such as non-melanoma skin cancer, post-transplant lymphoproliferative disorders, Kaposi sarcoma, anogenital cancer, lung cancer, renal cell carcinoma, or hepatocellular carcinoma [38].

Considering the results of the previously conducted studies, the effect of statins in reducing CRP levels in various diseases, with a significant impact on patient outcome, is of utmost importance. We aimed to explore this effect in the case of cardiac transplantation, representing a complex surgical procedure, with an even more detailed medical management. Therefore, the identification of factors that positively or negatively influence the outcome of a patient who underwent cardiac transplantation may be considered as one of the most important considerations in the research field. We considered CRP levels as a relevant indicator of inflammatory phenomena, thus we explored the potential connection between different medications and the levels of this inflammatory marker.

### Limitations

Our study encountered a number of limitations from a technical perspective. First, the retrospective character of the research bears in itself a number of limitations that have to be mentioned such as missing information, selection bias, and lack of patient compliance and presentation to follow-up hospital admission [39]. Another limitation was the small size of the cohort. Because of our country’s legislation with regard to the necessity of informed consent, difficulties to overcome paperwork, a lack of informational campaigns on the topic, and finding organs ready for transplantation pose challenges in the Romanian medical field [40]. Associated with the low number of individuals in our cohort, there is a possibility of false-positive results and an over-estimation of the results [41]. With regard to complications, undocumented, asymptomatic episodes of paroxysmal AFib may represent another important point of interest, while unobserved episodes could influence the results. Finally, patient adherence to the prescribed treatment is probably the most essential aspect of this study. Larger patient cohorts and further prospective studies are needed to increase the relevance and accuracy of our findings.

## 5. Conclusions

In summary, chronic preoperative administration of statins represent a protective factor against the development of 2-month postoperative complications of any cause in heart transplant recipients, simultaneously presenting an elevated risk for early-postoperative AKI. From the statin group, atorvastatin therapy had a higher risk of newly diagnosed postoperative T2DM. Second, we determined that CRP, TC, and LDL-c above the cut-off levels might be poor prognostic factors in the first 2 months after heart transplantation. Atorvastatin administration was associated with lower CRP values, underscoring the anti-inflammatory role of lipid-lowering therapy. Taking into account the small size of our cohort, further studies are required in order to make radical conclusions.

## Figures and Tables

**Figure 1 ijerph-20-03471-f001:**
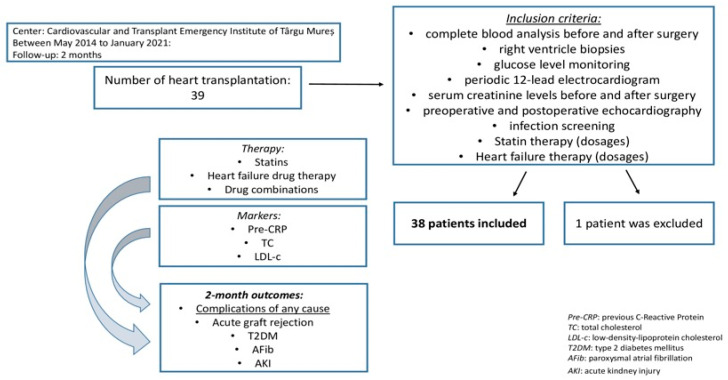
Inclusion criteria and study design.

**Figure 2 ijerph-20-03471-f002:**
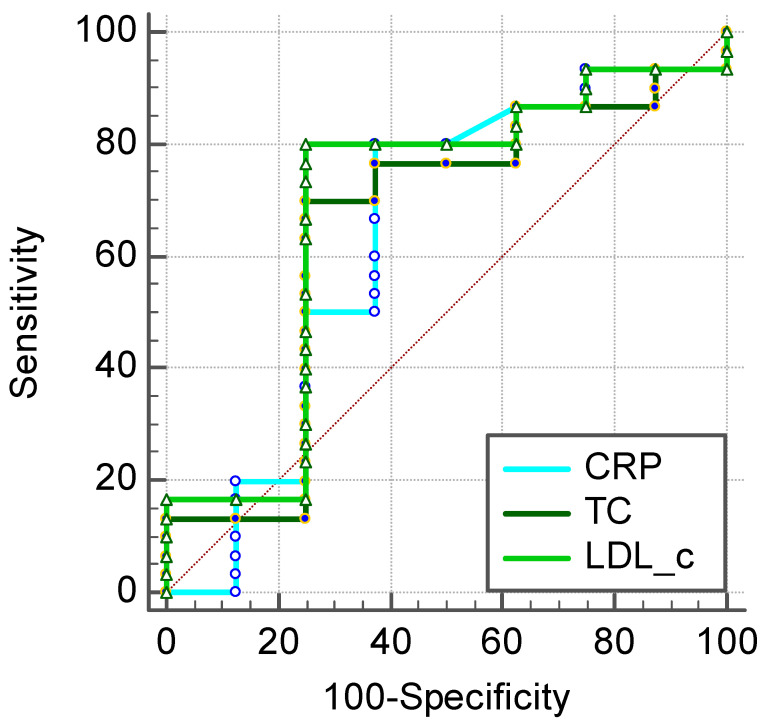
Prognostic values of CRP, TC, and LDL-c in the prediction of a 2-month complication of any cause after heart transplant.

**Table 1 ijerph-20-03471-t001:** Baseline patient characteristics.

	Total (*n* = 38)	Non-Statin (*n* = 27)	Statin (*n* = 11)	*p* Value
Age, years (Mean, SD)	41.21 (13.71)	38.15 (14.74)	48.73 (6.60)	0.02 *
BMI, kg/m^2^ (Mean, SD)	23.81 (5.18)	22.58 (5.48)	26.84 (2.68)	0.01 **
Pre-CRP, mg/L (Mean, SD)	9.3 (12.2)	10.8 (13.8)	5.5 (6.0)	0.27 **
Pre-LVEF, % (Mean, SD)	26.37 (13.10)	27.48 (15.28)	23.64 (4.03)	0.46 **
TC, mg/dL (Mean, SD)	165.87 (44.81)	164.37 (46.64)	169.55 (41.87)	0.75 *
LDL-c, mg/dL (Mean, SD)	106.05 (33.52)	107.22 (34.53)	103.18 (32.31)	0.74 *
HDL-c, mg/dL (Mean, SD)	36.47 (10.43)	34.85 (11.23)	40.45 (7.06)	0.12 **
non-HDl-c, mg/dL (Mean, SD)	129.39 (39.88)	129.52 (39.87)	129.09 (41.86)	0.97 *
VLDL-c, mg/dL (Mean, SD)	22.51 (10.66)	20.86 (10.23)	26.55 (11.10)	0.09 **
TG, mg/dL (Mean, SD)s	112.53 (53.32)	104.30 (51.16)	132.73 (55.51)	0.09 **
Castelli risk index I (Mean, SD)	4.78 (1.46)	4.97 (1.44)	4.33 (1.48)	0.24 **
Castelli risk index II (Mean, SD)	3.09 (1.19)	3.26 (1.20)	1.65 (1.10)	0.16 **
AI (Mean, SD)	3.78 (1.46)	3.97 (1.44)	3.33 (1.48)	0.24 **
AIP (Mean, SD)	0.47 (0.21)	0.46 (0.21)	0.49 (0.20)	0.70 *
Atorvastatin (*n*, %)	8 (21.1)	0 (0.0)	8 (72.7)	- ^†^
Rosuvastatin (*n*, %)	3 (7.9)	0 (0.0)	3 (27.3)	- ^†^
Spironolactone (*n*, %)	30 (78.9)	20 (74.1)	10 (90.9)	0.70 ^†^
Beta blockers (*n*, %)	30 (78.9)	19 (70.4)	11 (100.0)	- ^†^
Carvedilol (*n*, %)	27 (71.1)	18 (66.7)	9 (81.8)	0.72 ^†^
ACE-I (*n*, %)	17 (44.7)	10 (37.0)	7 (63.6)	0.53 ^†^
Ramipril (*n*, %)	12 (31.6)	6 (22.2)	6 (54.5)	0.54 ^†^
Pre-AFib (*n*, %)	8 (21.1)	5 (18.5)	3 (27.3)	0.15 ^†^
Pre-Complete LBBB (*n*,%)	6 (15.8)	2 (7.4)	4 (36.4)	- ^†^
Pre-CRT-D (*n*,%)	8 (21.1)	5 (18.5)	3 (27.3)	0.72 ^†^

BMI: body mass index; Pre-CRP: previous C-reactive protein; Pre-LVEF: previous left ventricle ejection fraction; TC: total cholesterol; LDL-c: low-density lipoprotein cholesterol; HDL-c: high-density lipoprotein cholesterol; non-HDL-c: non-high-density lipoprotein cholesterol; VLDL-c: very-low-density lipoprotein cholesterol; TG: triglycerides; AI: atherogenic index; AIP: atherogenic index of plasma; Pre-AFib: previous permanent AFib; Pre-complete LBBB: previous complete left bundle branch block; Pre-CRD-D: previous cardiac resynchronization therapy defibrillator implantation. * Unpaired *t* test; ** Mann–Whitney test; ^†^ Chi-square (χ^2^) test.

**Table 2 ijerph-20-03471-t002:** Presence of complications of any cause/specific type of complications based on the group distribution.

	Total (*n* = 38)	Non-Statin (*n* = 27)	Statin (*n* = 11)	*p* Value ^†^
Complications of any cause (*n*, %)	30 (78.9)	24 (88.9)	6 (54.5)	0.18
Acute graft rejection (*n*, %)	7 (18.4)	5 (18.5)	2 (18.2)	0.81
T2DM (*n*, %)	8 (21.1)	5 (18.5)	3 (27.3)	0.50
AFib (*n*, %)	6 (15.8)	5 (18.5)	1 (9.1)	0.09
AKI (*n*, %)	12 (31.6)	6 (22.2)	6 (54.5)	0.65

^†^ Chi-square (χ^2^) test.

**Table 3 ijerph-20-03471-t003:** Association between drug therapy and 2-month presence of postoperative complications of any cause/specific type of complications.

	Complications OR/95%CI	Acute Graft Rejection OR/95%CI	T2DM OR/95%CI	AFib OR/95%CI	AKI OR/95%CI
Statin	0.06 0.008–0.56 *p =* 0.0128	1.30 0.12–13.91	6.14 0.64–58.73	0.25 0.01–4.62	6.90 1.03–45.94 *p* = 0.0458
Atorvastatin	0.02 0.001–0.46 *p =* 0.0132	3.19 0.18–55.68	29.73 1.19–741.76 *p =* 0.0387	0.44 0.01–11.17	17.00 1.26–228.62 *p* = 0.0326
Spironolactone	- * -	0.88 0.10–7.57	0.31 0.03–2.45	- *-	5.45 0.49–59.97
Beta blocker	1.02 0.14–7.37	- * -	1.51 0.12–17.94	1.46 0.12–17.02	0.45 0.08–2.44
Carvedilol	1.20 0.20–6.87	2.17 0.21–22.38	3.12 0.30–32.25	0.67 0.09–5.09	0.50 0.10–2.37
ACE-I	1.41 0.25–7.80	1.96 0.32–11.80	1.42 0.24–8.38	0.18 0.01–1.80	1.61 0.36–7.12
Ramipril	1.12 0.17–7.32	1.72 0.27–10.72	3.06 0.51–18.06	0.31 0.02–3.27	1.30 0.27–6.30
Statin + Spironolactone	0.02 0.001–0.39 *p* = 0.0081	1.81 0.12–27.41	14.54 0.84–249.74	0.24 0.008–7.01	15.74 1.36–181.94 *p* = 0.0273
Atorvastatin + Spironolactone	0.02 0.001–0.46 *p* = 0.0132	3.19 0.18–55.68	29.73 1.19–741.76 *p* = 0.0387	0.44 0.01–11.17	17.00 1.26–228.62 *p* = 0.0326
Statin + Spironolactone + Carvedilol	0.02 0.001–0.46 *p* = 0.0132	3.19 0.18–55.68	29.73 1.19–741.76 *p* = 0.0387	0.44 0.01–11.17	17.00 1.26–228.62 *p* = 0.0326
Atorvastatin + Spironolactone + Carvedilol	- * -	3.98 0.18–84.42	- * -	0.640.02–19.23	11.580.77–173.63
Spironolactone + Carvedilol	0.71 0.13–3.79	0.97 0.17–5.32	1.33 0.24–7.16	1.52 0.23–9.71	1.01 0.24–4.20
Atorvastatin + Spironolactone + Ramipril	0.05 0.002–1.60	0.97 0.02–44.33	19.94 0.53–736.62	- * -	29.25 1.58–538.60 *p* = 0.0231
Spironolactone + Ramipril	0.67 0.09–4.77	1.03 0.12–8.35	5.55 0.84–36.46	0.37 0.03–4.21	1.94 0.37–10.04
Atorvastatin + Spironolactone + Carvedilol + Ramipril	- * -	0.73 0.02–24.16	- * -	0.93 0.04–17.49	9.25 0.65–130.86
Spironolactone + Carvedilol + Ramipril	0.47 0.06–3.57	1.16 0.14–9.41	7.19 1.04–49.34 *p* = 0.0446	0.42 0.03–5.08	1.25 0.22–7.18

* Cannot be calculated.

**Table 4 ijerph-20-03471-t004:** Area under the receiver operating characteristic (AUC ROC) curves of our 3 investigated markers: CRP, TC, and LDL-c.

	Cut-Off	AUC	Standard Error	95%CI	Sensitivity	Specificity
CRP (mg/L)	2.6	0.633	0.135	0.462–0.783	80.00	62.50
TC (mg/dL)	153	0.646	0.126	0.474–0.794	70.00	75.00
LDL-c (mg/dL)	91	0.683	0.124	0.513–0.824	80.00	75.00

**Table 5 ijerph-20-03471-t005:** Association between the ROC determined cut-off levels of preoperative CRP, TC, and LDL-c and the 2-month presence of postoperative complications/specific type of complications.

	Complications OR/95%CI	Acute Graft Rejection OR/95%CI	T2DM OR/95%CI	AFib OR/95%CI	AKI OR/95%CI
CRP > 2.6 mg/L	7.64 1.24–46.83 *p* = 0.0279	0.80 0.11–5.81	0.69 0,09–4.89	0.69 0.09–5.33	0.760.14–3.99
TC > 153 mg/dL	8.76 1.24–61.66 *p* = 0.0293	4.47 0.38–52.29	1.24 0.16–9.27	0.38 0.05–2.91	0.98 0.18–5.14
LDL-c > 91 mg/dL	11.25 1.54–82.15 *p* = 0.0169	3.00 0.22–40.81	1.61 0.14–17.69	3.42 0.19–59.41	0.55 0.09–3.07

**Table 6 ijerph-20-03471-t006:** Correlation/association between drug therapy and concentration levels of CRP.

Drug	CRP			
	r Value	95%CI Lower Limit	95%CI Upper Limit	χ^2^ Test *p* Value
Statin	−0.18	−0.48	0.15	0.23
Atorvastatin	−0.30	−0.57	0.02	0.03
Spironolactone	−0.22	−0.51	0.11	0.07
Carvedilol	−0.34	−0.60	0.01	0.45
Ramipril	−0.03	−0.36	0.29	0.71
Statin + Spironolactone	−0.22	−0.51	0.10	0.11
Atorvastatin + Spironolactone	−0.30	−0.57	0.02	0.03
Statin + Spironolactone + Carvedilol	−0.33	−0.59	0.008	0.03
Atorvastatin + Spironolactone + Carvedilol	−0.40	−0.64	0.08	0.01
Atorvastatin + Spironolactone + Ramipril	−0.16	−0.46	0.17	0.14

## Data Availability

The datasets generated and analyzed during the current study are available from the corresponding author on reasonable request.

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
