# Peer review of "The Role of Preoperative Chronic Statin Therapy in Heart Transplant Receipts—A Retrospective Single-Center Cohort Study"

_ijerph, 2023, doi:10.3390/ijerph20043471_

Round 1
Reviewer 1 Report
In this interesting paper, the authors investigated the role of pre-operative chronic statin therapy in heart transplantation recipients. They found that pre-operative chronic statin therapy works as a protective factor for the development of complications of any cause two months after transplantation. On the other hand, they found that patients treated with statin had a higher risk of acute kidney injury an d type 2 diabetes. I think this study could be interesting as the improvement of the survival of heart transplantation’s patients is crucial in the actual organ shortage scenario, but several things that could be improved:
1. The paper needs extensive English revision
2. When you compare baseline characteristics of patients (table 1) you have to include the p value of the comparison between the Statin group and the Non-statin one in order to understand if the differences you observed are statistically significant or not.
3. There is the same problem in table 2. It is useless to report the different percentages of the adverse events in the two groups without comparing it with the appropriate statistical analysis and reporting the p value.
4. In table 2 are reported 4 complications after heart transplantation. Does it mean you observed just these adverse events or there were others? Please clarify. Moreover, it is not clear what you mean by “complications” in general. Is it just the summa of the 4 adverse events in the table?
5. In table 3 there are reported the OR of different drugs and combination of drugs for the occurrence of different adverse events. I think the table should be simplified, removing most of the drugs associations and moreover I think the p value (that is reported in the text) should be included in the table as well.
6. In table 5 there is the same problem. Please include the p value
7. Results are not clearly presented in the tables.
8. I think it is confusing to focus both on statin in general and also on atorvastatin. In my opinion it would be better just to focus on one of them.
9. Why did you focus on 2 months’ postoperative complications? Of course the paper would me more valuable with a longer follow up, but if it is not possible why you didn’t just focus on in-hospital outcomes?
10. Limitations should be included in a dedicated paragraph.
Reviewer 2 Report
In this retrospective study, the authors investigated the relationship between preoperative chronic administration of statins to postoperative 2-month heart transplantation complications. Overall, 38 heart transplantations receipts were studied. In logistic regression, they found a statistical significance between Statin treatment and the presence of postoperative complications of any cause (OR: 0.06; p=0.0128) simultaneously presenting an elevated risk for early-postoperative acute kidney injury. From the Statin group, atorvastatin therapy had a higher risk of type 2 diabetes development (OR: 29.73, p=0.0387), and AKI (OR: 29.73; p=0.0387). C-Reactive protein, total cholesterol, and low-density lipoprotein cholesterol (LDL-c) resulted to be risk factors, atorvastatin administration being independently associated with lower CRP values. They concluded that chronic previous administration of statins was a protective factor for the development of 2-month postoperative complications of any cause in heart transplant receipts.
The study addresses an important issue related to the outcome of transplanted patients, although the low number of enrolled patients should suggest caution in interpreting results, as recognized by the authors. It would be of interest to present statin daily dosage and the safety profile of statin therapy since it is well-known that statin therapy may be offset by development of side effects.
Reviewer 3 Report
In the present paper Baba and colleagues provided a retrospective study aiming at dissecting possible relationship between Statin pre-operative treatment and post-operative 2-months hearth transplantation complications. The authors enrolled a study cohort consisting of 38 hearth transplanted patients ranging from 2014 and 2021 hospitalizations. Statistical analyses were performed to assess any association between Statin treatment and postoperative complications. Their data suggested that chronic preoperative administration of Statins represented a protective factor to postoperative “complications of any cause” 2-month following heart transplantation, albeit it simultaneously exposes to an elevated risk for early postoperative acute kidney injury.
Although the topic is of interest, many issues need to be properly addressed before considering the manuscript suitable for publication.
MAJOR ISSUES
1. The English should be revised. For example, the sentences at lines 50-54, 80-82, 179-180 are unclear and should be revised.
2. As concern the patient’s characteristics, I did not find indications about the gender of the patients. It has been widely acknowledged the role of gender in cardiovascular diseases.
3. Table 1: In table 1 the authors reported the pharmacologic treatments of both Non-Statin and Statin groups. In my opinion would be more informative if the authors could provide in table 1 the % of the patients recieving the therapy vs the number of the patients belonging to Non-statin or Statin Group. For example, Atorvastatin in Statin groups would be 72.72% (8/11)
4. Lines 165-166: the authors stated that “from 11 subjects receiving Statin therapy, 8 were on Atorvastatin and 2 on Rosuvastatin”. Firstly, what about the remaining 1 patient? Which treatment has he/she receive? Secondly, the authors should provide this information in table 1.
5. Given the subjects enrolled in the Statin and non-Statin groups have differences in pharmacological therapies, I did not understand the rationale behind the authors’ choice of assessing the associations between drug therapy and post-operative complications. For example, the most of Statin group patients received Atorvastatin, Spironolactone and Beta blocker (Table 1). why did the authors treat these therapies separately? In my opinion, the associations should be looked between the two groups and the post-operative complications.
6. Line 177: “Complications regardless of cause”: Does this item encompass the complications listed in table 2? It is not clear in the text.
7. Lines 182-183: Although the number is the same between the two groups, the % is very different. The authors must look at the % rather than the simple number given the differences in patient distribution within the groups.
8. The study cohort is too light to provide informative findings.
MINOR ISSUES
1. The acronym AKI should be provided at first mention in the text
Reviewer 4 Report
Baba et al investigated the relationship between preoperative chronic administrations of Statins to postoperative 2-month heart transplantation complications. They conclude that chronic previous administration of Statins represented a protective factor to the development of month's postoperative complications of any cause in heart transplant receipts. The topic is very interesting; however, there are several issues that should be addressed before the acceptance of this paper.
1. All abbreviations should be revised carefully and written for the first time before being used in the entire text by the authors. For example, in the abstract, the authors use the full name sometimes (line 34 Type 2 Diabetes Mellitus) and then in line 130 Type 2 Diabetes Mellitus (T2DM). The same thing in Line 134 and 135, mean and standard deviation, which should be abbreviated.
2. Line 117, 133, 196, there is space which should be corrected
3. New section: entitled "Inclusion and Exclusion Criteria and Experimental Design," it should be addressed in a separate subtitle and also illustrated in a diagram.
4. In the tables you could leave the average values to a single decimal.
5. In order to complete the lipid profile, Very low-density lipoprotein cholesterol, Anti-atherogenic index, Risk 1, and Risk 2 should be provided.
6. There are grammar errors throughout the article. The text should be proofread thoroughly for grammatical errors.

Round 2
Reviewer 1 Report
Dear authors,
thanks for answering all my questions. I think your paper now is suitable for publication.
Best regards
Reviewer 2 Report
The authors satisfactorily addressed the raised points.
Reviewer 3 Report
Please find in blue the replies to your answers
